# Strengthening TB care pathways: Integrating informal and AYUSH practitioners for early diagnosis and referral

Ridhima Sodhi[1]*, Shruti Goel[1], Rohit Baghel[1], Kiran Rade[2,3], Kshitij Khaparde[2], Harkesh Dabas[1], Shamim Mannan[1], Manoj Singh[1]

**1** William J Clinton Foundation, New Delhi, India, **2** World Health Organization, Country Office for India, New Delhi, India, **3** Stop TB Partnership, United Nations Office for Project Services (UNOPS), Geneva, Switzerland

* ridhimasodhi@gmail.com

## Abstract

Tuberculosis (TB) continues to be the leading cause of death from infectious diseases, disproportionately affecting populations in low- and middle-income countries, where diagnostic and financial barriers impede timely care. While alternative treatments may offer symptomatic relief, the cornerstone of TB management remains anti-tubercular therapy (ATT) as recommended by the World Health Organization (WHO). Yet, many individuals with TB-like symptoms first consult traditional healers and AYUSH (Ayurveda, Yoga, Naturopathy, Unani, Siddha, and Homeopathy) practitioners, resulting in delays in diagnosis and treatment initiation. This quasi-experimental study evaluated the impact of strengthening linkages between non-allopathic and allopathic providers, through formal referral systems, alongside the implementation of a free chest X-ray (CXR) screening system for presumptive TB patients. A mixed methods design found the significant role of friends and family in influencing care-seeking patterns, while a regression analysis within a Difference-in-Differences approach found the intervention to reduce diagnostic delay by over eight days (-8.6, 95% CI: −16.6, −0.5). The accelerated pathway to diagnosis was observed irrespective of which provider was first visited by the patient, holding promise for future such interventions, especially in high-burden regions, with significant reliance on informal providers.

## Introduction

Tuberculosis (TB) remains a major public health challenge, particularly in low- and middle-income countries, where economic burdens exacerbate the disease's impact [1–4]. Reliance on unqualified healthcare providers further delays diagnosis, increasing catastrophic expenditure and morbidity rates [2–4]. Dependence on informal providers is typically more common in rural areas [5], and studies indicate that more

**Data availability statement:** All of the data pertaining to this is available in a Github repository: https://github.com/ridhimasodhi/PatientPathway.

**Funding:** The baseline and endline assessments were funded by Wave 08 of the TB REACH program (Stop TB Partnership), awarded to the William J. Clinton Foundation. This grant supported the research and report preparation – but did not cover manuscript preparation. The Stop TB partnership had no involvement in study design, data collection, analysis, the decision to publish, or manuscript preparation. The authors received no additional funding for data compilation, statistical analysis, or manuscript preparation.

**Competing interests:** The authors have declared that no competing interests exist.

**Abbreviations:** ADITYA, Accelerating Diagnosis and Initiation of TB Treatment through your Action; CBNAAT, Cartridge-based Nucleic Acid Amplification Test; CI, Confidence interval; CME, Continuing Medical Education; CXR, Chest X-ray; DiD, Difference-in-differences; DxD, Diagnostic delay; FDC, Fixed Dose Combination; HCP, Health care provider; ICT, Information and Communication Technology; IDI, In-depth interview; IQR, Interquartile range; NAAT, Nucleic Acid Amplification Test; N-HCP, Number of health care providers consulted; OLS, Ordinary least squares; PD, Patient delay; TB, Tuberculosis; Xpert, Cartridge-based Nucleic Acid Amplification Test.

than 75% of primary care visits in India are made to them [5–7]. Provider shopping, and misdiagnosis, from unqualified as well as qualified providers, is common, and often results in prolonged delays, stretching for weeks or months before treatment begins [8–11]. Compounded further by non-specific antibiotic use and delay in diagnostic testing, these factors continue to be detrimental to TB elimination efforts within and outside India [8,12–18]. While significant provider-engagement efforts have been made [19–22], not many have integrated non-allopathic practitioners [2,4]. In Chhattisgarh, the region where this study is based, the breakdown of treated ailments by type of healthcare provider indicates that reliance on AYUSH (Ayurveda, Yoga & Naturopathy, Unani, Siddha, and Homeopathy) practitioners is 4.4% in rural areas and 1.4% in urban areas [23]. Systematic referral mechanisms, however, can be helpful in accelerating patient pathways, and help reduce catastrophic costs associated with such delays [2,24,25]. Set within Project ADITYA (Accelerating Diagnosis and Initiation of TB Treatment through your Action), the study evaluates the impact of formalizing referrals between non-allopathic and allopathic providers, whilst providing free diagnostic testing (Chest X-Ray or CXR; Nucleic Acid Amplification Test or NAAT) through an incentive and voucher-based model. Beyond analyzing TB patient pathways and their evolution during the intervention, this study provides a scalable roadmap for implementing similar strategies across other regions.

## Methods

### Drug sensitive TB Care services under Project Aditya

Project ADITYA was launched in February 2021 in Durg district, Chhattisgarh, as an inclusive private sector model for TB diagnosis and treatment, and was implemented through September 2022. Initiated with mapping all providers in the region, encompassing both qualified (AYUSH and MBBS/MD), and unqualified (those without a recognized medical degree) providers, alongside pharmacists, the project aimed to reduce diagnostic and treatment delays among DS-TB patients, through three components: a) voucher-based system providing free CXRs for presumptive TB patients through referrals to private laboratories, b) ensuring engaged providers can provide patients access to government funded/ free Xpert (Cartridge-cased NAAT) and fixed dose combination drugs (FDCs), and c) establishing an Information and Communication Technology (ICT) system for continuous patient management for TB patients diagnosed at facilities associated with Project Aditya. Engagement with both allopathic and non-allopathic providers was conducted through repeated interactions, including orientation meetings and continuing medical education (CME) sessions organized by the program team, and was supported by informal communication channels (e.g., WhatsApp groups) to facilitate information exchange. Of the more than 1,600 providers mapped during the intervention, 64% were engaged with the project, and 35% actively referred/notified patients. A total of 18 CMEs were organized during the intervention, which brought together all engaged providers, and were used to continually update providers on TB guidelines, project updates, and address any questions as well. The CXR referral incentive was set at a modest INR 100 (~ 1 USD), which was received by AYUSH and unqualified practitioners if a referred

patient got a confirmed TB diagnosis. Upon diagnosis, these providers could also serve as treatment supporters, receiving a sum of INR 500 at each treatment initiation and completion, totaling INR 1,000 or ~12 USD for each successfully treated patient.

### Study design

We used a mixed methods quasi-experimental intervention design, collecting quantitative and qualitative data concurrently [26,27]. The quantitative component employed a Difference-in-Differences (DiD) approach to assess the impact of the intervention on TB patient pathway, while thematic analysis of qualitative interviews examined how care seeking patterns evolved over time. The parallel trends assumption underlying the DiD framework is supported by the geographic and demographic uniformity of the study area across baseline and endline periods. Also, no other major TB interventions were introduced in the region during the study period, reducing the risk of time-varying confounding.

### Questionnaire design

A semi-structured questionnaire was used for in-depth interviews (IDIs) at baseline and endline. Lasting 45–120 minutes, the interviews assessed patient pathways, covering the journey from symptom onset to treatment completion. The endline assessment included additional questions for those engaged with Project Aditya, which included aspects of treatment support received during the treatment, an additional feature of the intervention. These questions were not analyzed as part of the assessment, as the latter focuses on patient pathway until the diagnosis/ treatment initiation.

### Data source

Three data sources were used. First, Ni-kshay, the Government of India's national TB surveillance portal, provided programmatic data used to sample TB patients for both baseline and endline assessments. Second, the Project ADITYA Management Information System (MIS) was used to identify project beneficiaries. This included both presumptive patients (who received CXR) and confirmed TB patients (who were further assessed through clinical and diagnostic assessment). Demographic details such as age, gender, and diagnosing district were extracted from these programmatic sources and later verified through patient interviews. Third, interview responses collected during the baseline and endline surveys provided detailed information on care-seeking patterns, diagnostic and treatment delays, and patient experiences. These datasets together informed both the quantitative and qualitative components of the analysis.

### Data selection (inclusion & exclusion criteria)

All adult (>=18 years) DS-TB patients, undergoing treatment in Durg, Chhattisgarh, were eligible. Individuals with drug-resistant or extrapulmonary forms of TB were excluded.

### Sampling method

Proportionate random sampling was used for baseline, ensuring representativeness across age and gender. The endline employed stratified proportionate random sampling, with strata defined by whether the patients were ADITYA beneficiaries. Type of provider at first contact was also considered for proportional representation.

### Sample size & sampling process

A total of 101 individuals were interviewed at baseline (34 women and 67 men) and 203 at endline. The larger endline sample reflected the inclusion of two strata - intervention and non-intervention groups. Sample size determination balanced pragmatic and scientific considerations: ensuring sufficient power to estimate intervention effects while capturing qualitative depth for thematic saturation [28–30].

Patients were pre-screened using a combination of Project MIS & Nikshay database, in order to ensure a) demographic representativeness, and b) confirm accurate tagging of patients across the intervention and control groups during the endline. Patients were telephonically contacted to obtain consent; approximately 60–70% declined, due to migration, work commitments, stigma, or the interview's duration. The denial rate reported for similar qualitative health studies, especially those involving disadvantaged or stigmatized populations, ranges from 30% to 89% [31,32]. No financial or material incentives were offered, which likely contributed to the latter. The final sample size is described in Table 1.

## Outcomes of interest

Two primary objectives were defined. First, to conduct a holistic assessment of the TB patient pathway, examining symptom onset, time to diagnosis, provider shopping, diagnostic testing, and associated expenditures, alongside understanding evolving patient and provider attitudes between the two time periods. The second objective was to quantitatively assess the intervention's impact on patient delay, number of providers visited prior to diagnosis, and diagnostic delay, independent of temporal, clinical and demographic factors. The outcomes of interest were grouped under two domains for ease of assessment:

• TB Patient Pathway: First point of contact for the individual (First POC), Patient delay (PD), First healthcare provider (First HCP), Primary reason for choosing provider, Number of providers consulted prior to diagnosis (N-HCP), and Diagnostic delay (DxD)

• Impact of intervention: Patient delay (PD), Number of providers visited (N-HCP), and Diagnostic delay (DxD)

Patient delay (PD) is defined as the number of days between symptom onset, and the first contact with an HCP. Diagnostic delay (DxD) is defined as the number of days between the first consultation with an HCP and confirmed diagnosis/ treatment initiation. Total delay is the sum of PD and DxD, and reflects the time from symptoms to diagnosis/ treatment initiation.

## Analysis

Qualitative interviews were coded thematically to capture patient experiences, barriers to care, and other pathway elements. Quantitative analysis was used to assess differences in key outcomes across intervention and non-intervention groups. Nonparametric tests (Kruskal Wallis rank-sum test) assessed statistical differences, while causal effects were estimated using DiD. Whenever presented, confidence intervals (CI) represent 95% interval values. For patient and diagnostic delay, Ordinary Least Squares (OLS) with robust standard errors were used. To explore distributional heterogeneity, quantile regressions were estimated at the median ($\tau = 0.5$) and upper quartile ($\tau = 0.75$). For the number of providers consulted, a Poisson model was used considering the non-negative and discrete nature of the variable. The standard

**Table 1. Sample size for baseline & endline groups (segregated by intervention status).**

|  | Baseline | Endline | |
|---|---|---|---|
|  | Non Intervention | Non Intervention | Intervention |
| **Provider Type** | **N (share)** | **N (share)** | **N (share)** |
| Ayush | 1 (1%) | 3 (4%) | 16 (13%) |
| Pharmacy | 23 (22.8%) | 1 (1%) | 17 (14%) |
| Qualified | 47 (46.5%) | 62 (78%) | 77 (63%) |
| Unqualified | 30 (29.7%) | 14 (18%) | 13 (11%) |
| **TOTAL** | **101** | **80** | **123** |

modeling specification is provided in the supplementary text. Covariate selection was done basis a combination of theoretical relevance and prior literature on TB care-seeking behaviour. A series of models were fitted, sequentially adding demographic characteristics (age, gender, type of household, occupation), the first point of contact, first provider type, loss of income or/and job, symptom severity, comorbidities, TB history of individual/family, and primary drivers for choosing a provider. Best fit was measured by adjusted R Squared.

### Robustness checks

To test the validity of our findings across the outcome distribution, particularly given the positively skewed nature of delay variables, quantile regressions at the 50th and 75th percentiles were conducted. Sparse categories (fewer than 10 observations) for *first point of contact* and *occupation* were combined to address convergence issues. Koenker and Machado's pseudo R-squared values are reported for these models.

Quantile treatment effect (QTE) plots across the 10th to 90th percentiles ($\tau = 0.1$ to $0.9$) are provided in the supplementary information to illustrate intervention's impact across the distribution of patient delay [Fig A in S1 Appendix]. Potential overfitting and multicollinearity were assessed using the Generalized Variance Inflation Factors (GVIF), all of which were well below the accepted threshold of two [Tables A-C in S1 Appendix]. Additional sensitivity checks included analyzing the impact of different provider types on diagnostic delay [Table D in S1 Appendix], as well as number of providers visited [Table E in S1 Appendix].

### Software

Audio recordings were manually transcribed in Excel and organised by thematic categories. The cleaned data were stored as CSV files for quantitative and qualitative analysis. ATLAS.ti was used for thematic coding, and RStudio (R version 2022.07.01) for statistical analysis and visualisation, using the packages dplyr, tidyr, gtsummary, and ggplot2.

### Data consolidation and confidentiality

Audio recordings were stored on a password-protected hard drive accessible only to the core research team. Transcriptions were anonymised by removing identifiable information, and each participant was assigned a unique identification code prior to analysis.

### Ethical considerations

Institutional ethics clearance was obtained for both study phases. The baseline received clearance from the Sigma Institutional Review Board, New Delhi (Ref. No. 10052/IRB/20–21; 22 January 2021), and the endline from the TRIOs Institutional Review Board, New Delhi (Ref. No. 101/2022/013/I/CHAI; 15 December 2022).

Recruitment for the baseline occurred between 17 February and 10 March 2021, and for the endline between 1 January and 1 February 2023. Informed consent was sought for all interactions between the patient and researcher, and no minors were interviewed.

## Results

### Demographic summary

A total of 101 and 203 DS-TB patients were interviewed during the baseline and endline phases, wherein the latter were further segregated by intervention (123) and non-intervention groups (80) [Table 2]. Majority of patients (>61%) interviewed were male across the phases, with mean age among the baseline cohort to be significantly lower (37 years vs 41 years and 45 years for control and endline groups, respectively; p = 0.003). The share of unemployment across phases varied between 40% to 48%. Patients with reported TB history in self or family varied between 18% and 21%. The endline

**Table 2. Summary statistics for dataset for individuals interviewed in baseline & endline.**

| Characteristic | baseline N = 101[1] | endline - Control N = 80[1] | endline - intervention N = 123[1] | p-value[2] |
|---|---|---|---|---|
| **Intervention** | 0 (0%) | 0 (0%) | 123 (100%) | <0.001*** |
| **Males** | 67 (66%) | 49 (61%) | 83 (67%) | 0.6 |
| **Age** | | | | 0.003** |
| Median (Q1, Q3) | 33 (27, 46) | 38 (28, 54) | 45 (30, 55) | |
| Mean | 37 | 41 | 45 | |
| **Occupation** | | | | 0.021* |
| Agricultural/Labor | 46 (46%) | 22 (28%) | 27 (22%) | |
| govt | 3 (3.0%) | 4 (5.0%) | 5 (4.1%) | |
| private | 12 (12%) | 9 (11%) | 22 (18%) | |
| Retired/Unemployed | 40 (40%) | 37 (46%) | 59 (48%) | |
| Self Employed | 0 (0%) | 8 (10%) | 10 (8.1%) | |
| **Type of HH (joint = 1)** | 54 (53%) | 33 (41%) | 55 (45%) | 0.2 |
| **Symptom Severity** | | | | 0.007** |
| mild/moderate | 75 (74%) | 46 (58%) | 67 (54%) | |
| severe | 26 (26%) | 34 (43%) | 56 (46%) | |
| **Comorbidity** | 58 (57%) | 27 (34%) | 33 (27%) | <0.001*** |
| **Distress Selling/borrowing** | 21 (21%) | 16 (20%) | 25 (20%) | >0.9 |
| **Loss of livelihood** | 25 (25%) | 4 (5.0%) | 14 (11%) | <0.001*** |
| **TB History** | 21 (21%) | 18 (23%) | 22 (18%) | 0.7 |
| **First POC** | | | | 0.009** |
| Ayush | 2 (2.0%) | 1 (1.3%) | 3 (2.4%) | |
| Community Health Worker | 6 (5.9%) | 2 (2.5%) | 1 (0.8%) | |
| Friends/Family | 85 (84%) | 58 (73%) | 102 (83%) | |
| Pharmacy | 3 (3.0%) | 0 (0%) | 1 (0.8%) | |
| Qualified | 0 (0%) | 17 (21%) | 12 (9.8%) | |
| Unqualified | 5 (5.0%) | 2 (2.5%) | 4 (3.3%) | |
| **First Doc** | | | | <0.001*** |
| Allopathic | 47 (47%) | 62 (78%) | 77 (63%) | |
| Ayush | 1 (1.0%) | 3 (3.8%) | 16 (13%) | |
| Self/pharmacy | 23 (23%) | 1 (1.3%) | 17 (14%) | |
| Unqualified | 30 (30%) | 14 (18%) | 13 (11%) | |
| **Patient Delay (PD)** | | | | <0.001*** |
| Median (Q1, Q3) | 35 (30, 50) | 10 (7, 22) | 10 (7, 17) | |
| Mean | 42 | 18 | 18 | |
| **Diagnostic Delay (DxD)** | | | | <0.001*** |
| Median (Q1, Q3) | 27 (20, 34) | 4 (1, 15) | 3 (1, 8) | |
| Mean | 32 | 17 | 10 | |
| **Total Delay** | | | | <0.001*** |
| Median (Q1, Q3) | 62 (54, 84) | 19 (10, 46) | 15 (11, 30) | |
| Mean | 74 | 35 | 28 | |
| **Primary Reason for choosing provider** | | | | 0.8 |
| Affordability | 26 (26%) | 6 (7.5%) | 5 (4.1%) | |
| Known/Family Networks | 11 (11%) | 45 (56%) | 68 (55%) | |
| Others | 7 (6.9%) | 2 (2.5%) | 5 (4.1%) | |
| Proximity | 43 (43%) | 7 (8.8%) | 15 (12%) | |

*(Continued)*

**Table 2.** (Continued)

| Characteristic | baseline<br>N = 101[1] | endline - Control<br>N = 80[1] | endline - intervention<br>N = 123[1] | p-value[2] |
|---|---|---|---|---|
| Referral | 14 (14%) | 20 (25%) | 30 (24%) | |
| **Total providers consulted** | | | | <0.001*** |
| 1 | 12 (12%) | 56 (70%) | 73 (59%) | |
| 2 | 9 (8.9%) | 24 (30%) | 48 (39%) | |
| 3 | 65 (64%) | 0 (0%) | 2 (1.6%) | |
| 4 | 15 (15%) | 0 (0%) | 0 (0%) | |
| Median (Q1, Q3) | 3.00 (3.00, 3.00) | 1.00 (1.00, 2.00) | 1.00 (1.00, 2.00) | |
| Mean | 2.82 | 1.3 | 1.42 | |

[1]n (%); *p < 0.05; **p < 0.01; ***p < 0.001.

Notes:

[2]Values represent number of individuals/ TB patients interviewed, along with % in brackets; n(%).

[3]p value represents values from Kruskal-Wallis rank sum test.

cohort had a significantly higher share of individuals with severe symptoms (43%-46% vs 26%, p = 0.007), but lower share with comorbidities (27%-34% vs 57%, p < 0.001). While distress sales due to TB-related expenditures were similar across phases (20%-21%), the share of those losing jobs were higher at baseline (25%) than at endline-control (5%) and endline-intervention (11%).

## TB patient pathway

**First point of contact (First POC).** Majority of TB patients (73% - 84%) reported sharing symptoms with friends and family [Table 2]. Other points of contact include community healthcare workers (CHWs), pharmacists, village doctors, and AYUSH practitioners. Endline assessment included an additional probe asking if individuals went directly to allopathic providers, to which 10% and 21% of individuals responded in admission from the intervention and control group, respectively. Such a probe was not present for the baseline assessment.

**Patient delay (PD).** Individuals from the endline cohort reportedly took 10 median days to seek medical consultation from the onset of their symptoms, relative to 35 median days taken by the baseline cohort [Table 2, Fig 1].

**First healthcare provider (First HCP).** Individuals from the endline cohorts were significantly more likely to seek consultation from an allopathic provider (78% and 63% for the control and intervention groups, respectively), relative to those in the baseline cohort (47%). Subsequently, those from endline were less likely to seek consultation from pharmacy, as well as unqualified providers. The share of those consulting with AYUSH were highest from the endline-intervention group (13%) [Table 2].

**Drivers behind choosing provider.** Primary factors influencing provider choice include affordability, family networks, proximity, and doctor referrals. Reliance on proximity decreased between phases – from 43% during baseline to 8.8% and 12% for endline control and intervention groups, respectively [Table 2]. Reliance on referrals, on the other hand, increased from 14% (baseline) to 24%-25% (endline). Affordability was a key concern for the baseline group (26%), relative to those from the endline group (7.5% and 4.1% for control and intervention groups, respectively).

**Number of total providers visited (N-HCP).** The number of total providers visited before diagnosis decreased from 3 median providers at baseline to 1 median provider at endline [Table 2]. For both the control and intervention cohorts of endline, most patients (>=98%) were diagnosed after visiting one or two providers, relative to less than 20% at baseline [Table 2].

**Diagnostic delay (DxD).** Days between the first provider visit and TB diagnosis reduced significantly between baseline (Median = 27, Mean = 32) and endline (Median = 3, Mean = 10 for control; Median = 4, Mean = 17 for intervention) phases

PLOS Global Public
Health

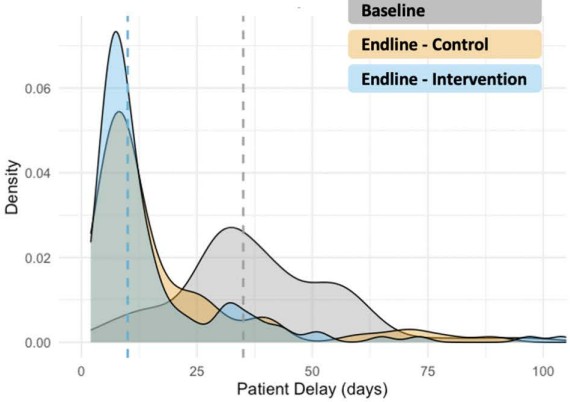 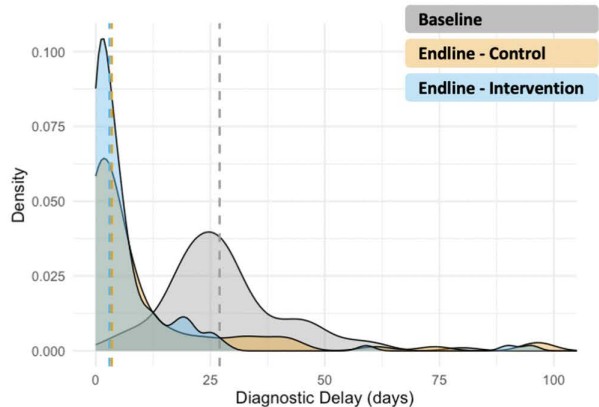

**Fig 1. Patient Delay & Diagnostic Delay – Density Plots, segregated by intervention cohorts.** *Note: Dotted lines represent cohort medians, x-axis censored beyond 100th day for visual clarity. Negligible data lost (6 and 7 points lost for patient and diagnostic delay, respectively).*

[Table 2]. Mean diagnostic delay typically increased with the number of providers, but the variation was better explained by the assessment periods [Table 3, Fig 2]. Median values remained relatively stable, while mean values shifted more sharply, suggesting that the change in delay was driven by extreme values, rather than the middle percentiles.

### Impact of intervention (Difference in Differences - DiD)

DiD was used to estimate the impact of the intervention on three variables – patient delay, number of providers visited prior to diagnosis, and diagnostic delay. A series of regression models were fitted before deciding on the models presented, decided based on a combination of model parsimony and adjusted R-squared values.

**Patient delay (PD).** The intervention is not observed to have any impact on patient delay, a finding consistent with OLS and quantile regressions [Table 4, Fig A in S1 Appendix]. The decline in patient delay between baseline and endline is estimated to be driven significantly by time factors (-19.4 days, C.I.: -26.8, -12.0), with consistent effects at the median (-23.9 days) and 75th percentile (-17.9 days). Comorbidities are observed to significantly reduce patient delay by about 5 days, an effect consistent at the median (3.6 days), but not at the upper (75th) quartile of the distribution. Symptom severity, on the other hand, is estimated to decrease patient delay by 5 days, but with a lower effect at the median (2.2 days), which goes insignificant at the upper quartile. Choosing providers based on proximity was linked to increase delay by more than a week (9 days, CI: -0.5, 18.3), but the association was not significant at the median and 75th percentile.

**Number of providers visited (N-HCP).** While the model estimates the intervention to have negligible impact on the number of providers visited (IRR = 1.02, CI: 0.77, 1.27), temporal effects alone explain about 50% decline in providers

**Table 3. Median & Mean Diagnostic Delay, segregated by number of doctors visited and intervention cohorts.**

| | Median Diagnostic Delay | | | Mean Diagnostic Delay | | |
|---|---|---|---|---|---|---|
| | Baseline | Endline | | Baseline | Endline | |
| Number of Docs Visited | | Control | Intervention | | Control | Intervention |
| 1 | 25.5 | 3.5 | 3 | 23.2 | 14.2 | 5.4 |
| 2 | 16 | 4 | 2.5 | 23.0 | 23.1 | 18.1 |
| 3 | 27 | | 5.5 | 35.5 | | 5.5 |
| 4 | 27 | | | 29.0 | | |

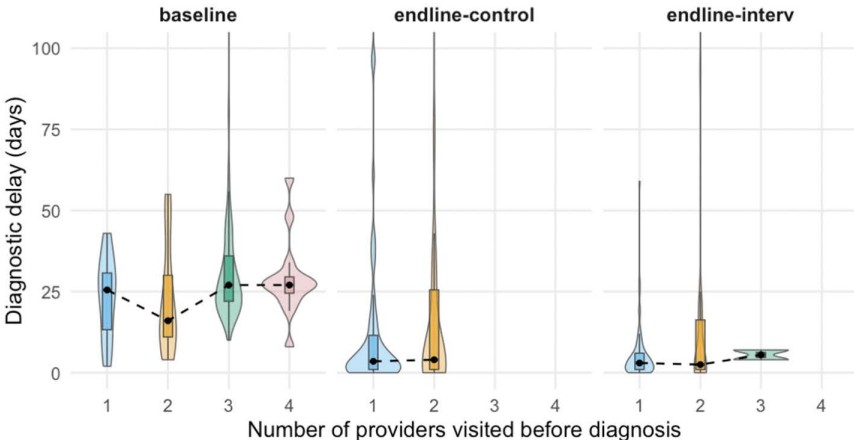

**Fig 2. Violin Plots - Diagnostic Delay, segregated by number of providers and intervention cohorts.** *Note: The dashed line connects group medians.*

consulted (IRR = 0.49, CI: 0.23, 0.75) [Table 5]. Compared with allopathic providers, patients who first sought care from AYUSH providers (IRR = 1.68; CI: 1.16–2.36), pharmacies/self-medication (IRR = 1.46; CI: 1.14–1.86), or unqualified providers (IRR = 1.48; CI: 1.18–1.85) are estimated to have significantly higher provider visits prior to diagnosis.

**Diagnostic delay (DxD).** The intervention is expected to significantly reduce diagnostic delay by more than 8 days (-8.6, CI: -16.6, -0.5), but the effect is not significant at the median or the 75th percentile [Table 6]. Time effects account for a significant majority of the reduction (-11.2 days, CI: -21.7, -0.7), and the effects stay consistent at the median (−21.00 days) and upper quantile (-17.6 days) [Table 6]. History of TB in patient or family was associated with a slight reduction in diagnostic delay, but no other covariate showed consistent association. The OLS model predicted diagnostic delay to be consistently shorter in the intervention group, across all provider types [Fig 3], a finding consistent with another model specification with an interaction term on provider type [Table D in S1 Appendix].

## Discussion

To our knowledge, this is the first quasi-experimental mixed-method study to assess the impact of an all-inclusive provider intervention on the TB patient pathway. By reflecting real-world care-seeking behaviour, where many patients do not initially consult MBBS/MD-qualified doctors, the analysis finds evidence underscoring the impact of incentivized referral mechanisms between providers on reducing diagnostic delay associated with preference for alternative and unqualified doctors, provider shopping, and lack of access to diagnostic services. The findings also highlight the significant influence of friends and family in care-seeking patterns, reinforcing the need for family centered and community-based approaches for TB elimination [Fig 4, Table 7].

### TB patient pathway

**First point of contact.** Social networks and personal beliefs have historically been found to account for a majority (42%) of the care-seeking behaviour [33]. Advances in telemedicine may influence this in times to come, but our results as well as prior literature find that individuals typically discuss their symptoms first with a trusted friend or a family member, and not a doctor [34,35]. If at all a faster or initial contact is made with a qualified professional, the same is typically linked to proximity with the doctor, or linkage through employment-linked facilities [36–38].

**Table 4.** OLS & Quantile Regression, investigating impact of intervention on Patient Delay. Values displayed are estimated coefficients, along with 95% C.I.

| | OLS (Robust SE) | Robustness tests | |
| --- | --- | --- | --- |
| | | Quantile (0.5) (Robust SE) | Quantile (0.75) (Robust SE) |
| Intervention | 0.4 | -0.2 | -5.1 |
| | (-6.3, 7.0) | (-3.4, 3.0) | (-14.1, 3.9) |
| Time | -19.4*** | -23.9*** | -17.9*** |
| | (-26.8, -12.0) | (-28.9, -18.9) | (-30.9, -4.9) |
| Age | 0.1 | 0.03 | 0.05 |
| | (-0.1, 0.3) | (-0.05, 0.1) | (-0.2, 0.3) |
| Gender: Male | 1.3 | -0.9 | -0.6 |
| | (-3.9, 6.5) | (-4.2, 2.5) | (-7.9, 6.8) |
| Occupation: Government | -7.7* | 0.2 | -2.7 |
| | (-17.0, 1.5) | (-7.7, 8.2) | (-20.0, 14.6) |
| Occupation: Private | 2.6 | -0.5 | 2.5 |
| | (-6.6, 11.8) | (-5.4, 4.3) | (-12.4, 17.5) |
| Occupation: Retired/Unemployed | -0.2 | -0.02 | 0.8 |
| | (-6.1, 5.7) | (-3.0, 2.9) | (-8.2, 9.7) |
| Occupation: Self Employed | 8.9 | 3.2 | 11.8 |
| | (-2.3, 20.2) | (-7.1, 13.6) | (-11.9, 35.5) |
| Type Household: Joint | 2.4 | 0.3 | 5.2 |
| | (-3.1, 7.9) | (-2.7, 3.3) | (-2.7, 13.0) |
| First POC: Friends/Family | 5.7 | 0.8 | 0.4 |
| | (-4.2, 15.6) | (-7.4, 8.9) | (-11.3, 12.1) |
| First POC: Qualified Doctor | 6.3 | -0.2 | -1.8 |
| | (-8.9, 21.5) | (-10.0, 9.7) | (-19.6, 15.9) |
| First POC: Unqualified Doctor | 4.8 | 12.3** | -2.4 |
| | (-11.0, 20.6) | (1.3, 23.2) | (-23.8, 19.0) |
| Comorbidities: True | 4.8* | 3.6* | 3.8 |
| | (-0.5, 10.1) | (-0.2, 7.4) | (-5.1, 12.7) |
| TB History (Self/Family): True | 2.4 | -0.2 | 5.4 |
| | (-4.2, 8.9) | (-3.4, 2.9) | (-7.6, 18.4) |
| Symptom Severity: Severe | -5.1** | -2.2* | -4.9 |
| | (-9.8, -0.4) | (-4.8, 0.4) | (-12.6, 2.8) |
| Prov Choosing Reason: Known/Family Networks | 4.3 | 0.6 | 0.3 |
| | (-3.7, 12.2) | (-5.2, 6.5) | (-12.8, 13.3) |
| Prov Choosing Reason: Others | 4.1 | 0.9 | 3.4 |
| | (-5.5, 13.7) | (-7.0, 8.8) | (-13.5, 20.4) |
| Prov Choosing Reason: Proximity | 8.9* | 3.6 | 11.4 |
| | (-0.5, 18.3) | (-3.8, 10.9) | (-5.0, 27.9) |
| Prov Choosing Reason: Referral | -0.5 | -0.6 | -0.4 |
| | (-8.6, 7.6) | (-6.8, 5.7) | (-12.9, 12.0) |
| # Symptoms | -1.3 | -0.2 | -0.9 |
| | (-3.6, 0.9) | (-1.4, 1.0) | (-3.5, 1.7) |
| Constant | 25.9*** | 32.1*** | 37.2*** |
| | (14.6, 37.1) | (21.1, 43.2) | (16.9, 57.4) |

*(Continued)*

**Table 4.** (Continued)

| | OLS (Robust SE) | Robustness tests | |
| --- | --- | --- | --- |
| | | Quantile (0.5) (Robust SE) | Quantile (0.75) (Robust SE) |
| **Observations** | 304 | 304 | 304 |
| **R2** | 0.3 | | |
| **Adjusted R2** | 0.2 | | |
| **Pseudo R-squared** | | 0.304 | 0.212 |
| **Residual Std. Error** | 22.294 (df = 283) | | |
| **F Statistic** | 5.101*** (df = 20; 283) | | |

Note:*p < 0.1; **p < 0.05; ***p < 0.01;

1. There was 1 retired individual, who was clubbed with unemployed.

2. First Point of Contact => we clubbed AYUSH (N = 6), CHW or Community Healthcare Workers (N = 9), and Pharmacies (N = 4), under one category.

**Table 5.** Poisson Regression, investigating impact of intervention on number of providers visited by the individual. Values displayed are Exponentiated Coefficients or Incidence Rate Ratios (IRRs), along with 95% C.I.

| | IRR + Robust SE |
| --- | --- |
| Intervention | 1.02 (0.77, 1.27) |
| Time | 0.53*** (0.26, 0.80) |
| First Doc Type: AYUSH | 1.68*** (1.32, 2.03) |
| First Doc Type: Self/Pharmacy | 1.46*** (1.21, 1.70) |
| First Doc Type: Unqualified | 1.48*** (1.26, 1.70) |
| Age | 1.00 (0.99, 1.01) |
| Gender: Male | 0.97 (0.77, 1.18) |
| Type Household: Joint | 0.93* (0.76, 1.11) |
| Occupation: Government | 0.99 (0.52, 1.47) |
| Occupation: Private | 1.03 (0.75, 1.31) |
| Occupation: Retired/Unemployed | 0.94 (0.72, 1.16) |
| Occupation: Self Employed | 1.00 (0.57, 1.43) |
| Symptom Severity: Severe | 0.93* (0.74, 1.12) |
| Comorbidities: True | 1.03 (0.82, 1.23) |
| TB History (Self/Family): True | 0.94 (0.72, 1.16) |
| Prov Choosing Reason: Known/Family Networks | 0.90 (0.59, 1.22) |
| Prov Choosing Reason: Others | 0.95 (0.50, 1.40) |
| Prov Choosing Reason: Proximity | 0.90 (0.62, 1.18) |
| Prov Choosing Reason: Referral | 0.99 (0.67, 1.32) |
| **Constant** | 2.56*** (2.16, 2.96) |
| **Observations** | 304 |
| **Akaike Inf. Crit.** | 820.62 |

Note:*p < 0.1; **p < 0.05; ***p < 0.01;

1. There was 1 retired individual, who was clubbed with unemployed.

**Table 6. OLS & Quantile Regression, investigating impact of intervention on Diagnostic Delay. Values displayed are estimated coefficients, along with 95% C.I.**

| | | Robustness tests | |
| --- | --- | --- | --- |
| | OLS (Robust SE) | Quantile (0.5) (Robust SE) | Quantile (0.75) (Robust SE) |
| Intervention | -8.6** | 0 | -4.8 |
| | (-16.6, -0.5) | (-2.0, 2.0) | (-15.9, 6.3) |
| Time | -11.2** | -21.0*** | -17.6** |
| | (-21.7, -0.7) | (-24.6, -17.4) | (-31.0, -4.2) |
| Age | 0.1 | 0 | 0 |
| | (-0.2, 0.4) | (-0.1, 0.1) | (-0.2, 0.2) |
| Gender: Male | -0.9 | 1 | -0.6 |
| | (-7.5, 5.8) | (-0.9, 2.9) | (-6.9, 5.7) |
| Type Household: Joint | 0.2 | 0 | -2.4 |
| | (-6.6, 7.1) | (-1.9, 1.9) | (-7.0, 2.2 |
| First Doc Type: AYUSH | 10.8 | 0 | 14 |
| | (-5.3, 26.8) | (-9.6, 9.6) | (-12.3, 40.3) |
| First Doc Type: Self/Pharmacy | 11.4 | 3 | 4 |
| | (-3.2, 26.0) | (-0.8, 6.8) | (-4.5, 12.5) |
| First Doc Type: Unqualified | 7.8 | 2 | 6.2 |
| | (-3.1, 18.6) | (-1.9, 5.9) | (-2.9, 15.3) |
| Comorbidities: True | 6 | 0 | 4.8 |
| | (-1.2, 13.1) | (-2.4, 2.4) | (-3.7, 13.3) |
| Symptom Severity: Severe | 1.5 | 0 | 4.2 |
| | (-5.1, 8.1) | (-2.0, 2.0) | (-0.9, 9.3) |
| TB History (Self/Family): True | -2.6 | -2.0* | -3.6 |
| | (-10.0, 4.7) | (-4.2, 0.2) | (-9.8, 2.6) |
| Constant | 20.0** | 24.0*** | 29.4*** |
| | (3.0, 37.0) | (19.3, 28.7) | (16.8, 42.0) |
| Observations | 304 | 304 | 304 |
| R2 | 0.12 | | |
| Adjusted R2 | 0.09 | | |
| Pseudo R-squared | | 0.299 | 0.131 |
| Residual Std. Error | 31.21 (df = 292) | | |
| F Statistic | 3.63*** (df = 11; 292) | | |

Note: *p < 0.1; **p < 0.05; ***p < 0.01;

1. There was 1 retired individual, who was clubbed with unemployed.

2. First Point of Contact => we clubbed AYUSH (N = 6), CHW or Community Healthcare Workers (N = 9), and Pharmacies (N = 4), under one category.

**Patient delay (onset of symptoms to seeking professional care).** In the National Tuberculosis Prevalence Survey 2019–2021, it was observed that up to 64% of those with presumptive TB symptoms or signs did not immediately seek care [39]. Prior research finds that individuals are likely to visit a medical professional (or not) based on advice given by close contacts, but prefer seeking a professional opinion in case of conflicting guidance by different friends and family members [33–35]. Community-based approaches encouraging a consultation with medical professional are hence, likely to have a positive effect on timely care-seeking [33,34,39]. Median patient delay in our study was just above a

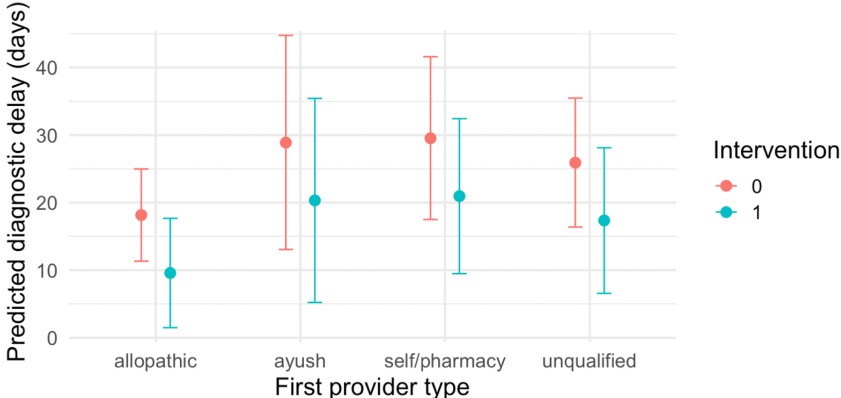

**Fig 3. Predicted Diagnostic Delay – Segregated by provider types and intervention status.**

**Primary factors affecting patient delay**

Home
Occupation Remedies    Gender
Barriers

Proximity to a
health facility    Symptom
Severity

No measurable effect from the intervention.
Reduced by ~20 days due to temporal effects, and post-COVID
normalization of healthcare service delivery

**Patient Delay**
*(time between onset of symptoms and seeking professional help)*

**Stage 1**
**Experiencing Symptoms**

cough
lethargy
fever    Body ache

**Stage 2**
**Sharing with someone**

Family members
(~80%)    Pharmacists /
Home Remedies
(>=23%)

Community
Health Workers;
<10%

**Advice Received**

Check with a
medical
provider    Incorrect self diagnosis –
confusion with cold / fever /
viral

Home
Remedies
(herbal drink,
taking rest)    Suggestion to visit nearby
doctors (could be quack /
AYUSH / government or
private doctor)

Referrals to a qualified doctor

**Increased post / during the intervention period**

**Stage 3**
**Meeting a medical practitioner**

**Stage 4**
**Doctor Shopping / Waiting for relief**

1) Following advice / Misdiagnosis / Not getting relief
2) Visiting additional doctors (typically, 1 to 3 doctors are visited)

*Referral mechanisms under the Aditya intervention accelerated this process, making "engaged" village doctors and AYUSH practitioners more likely to refer patients to a qualified doctor*

**Stage 5**
**Correct Diagnosis**

**Choice of the type of doctor is driven by several factors**

Referrals by known / village doctors

Affordability

Known / Family Doctor

Proximity

Government facilities preferred by government servants / families of those

Allopathic
(47% => increased to 78%)

AYUSH (1% - 3%)

Unqualified (30% => reduced to 11%)

**Diagnostic Delay**
*(time between first patient visit and actual diagnosis)*

Estimated to have reduced by more than a week due to the intervention, as well as additional 11 days due to health system improvements over the time-period

**Fig 4. *Typical TB Patient Pathway and its evolution under Project Aditya.* Note**: The figures given against certain factors (such as share of AYUSH doctors visited) are taken from Tables 2–5, and are approximated.

**Table 7. Triangulating Qualitative and Quantitative Findings Across the TB Patient Pathway.**

| Stage | Theme | Quantitative Finding | Qualitative Insight |
|---|---|---|---|
| Stage 1 | Symptom Onset | Severe symptoms associated with shorter patient delay (OLS: −5 days); effect significant at median, but not 75th percentile | Symptom severity influences care-seeking. Individuals with more severe symptoms report shorter delays, though this effect is attenuated at the upper quantile. The latter reflects an even stronger role of community and ease of access to medical care in breaking these patterns<br>*"I drank cough syrup & paracetamol for many days to reduce my cough & fever"* |
| Stage 1 | Existing comorbidities | Comorbidity increased patient delay (~4.8 days in OLS; significant at median, but not at 75th percentile) | Individuals with pre-existing conditions may attribute new symptoms to known illnesses, delaying TB-specific care. This effect is concentrated below the upper quartile |
| Stage 2 | First Contact | 3 in 4 patients reported that they first consult with family/friends | Most individuals first consult family or friends. While this can delay clinical care, it also suggests potential to leverage community networks for faster referrals<br>*"Some herbal remedies given by family members."* |
| Stage 3 | First Provider | First visiting AYUSH/pharmacy/unqualified providers increases provider count by 50%-70%; relative to visiting an allopathic provider | Initial visits to unqualified providers are common and associated with longer diagnostic delays and more provider visits<br>*"My jholachap doctor advised me to visit that doctor..."* |
| Stage 4 | Waiting for symptom relief/ Provider Shopping | Number of providers visited did not significantly reduce due to intervention, even as provider referrals increased – from 14% to 25% between baseline and endline | Misdiagnosis and symptom-based treatment is common, leading to repeated provider visits. Provider referrals increased during intervention, possibly contributing to shorter pathways<br>*"Vaid doctor prescribed her medicine for two weeks for Typhoid. She went twice there. But she did not get better… decided to consult another doctor."* |
| Stage 5 | Diagnosis & Referral | Diagnostic delay significantly reduced during endline period; shorter DxD associated with TB history and formal referral mechanisms | Early referral to formal providers can accelerate diagnosis, especially if preceded by appropriate testing<br>*"After receiving the reports … the doctor referred the patient to the hospital …"* |

month (Median = 35, Mean = 42), aligning with existing estimates [8,40]. This is primarily linked to initial misattribution of symptoms to minor ailments such as seasonal flu or cough, and limited awareness of TB symptoms. Similar behavioral patterns, including perceiving symptoms as non-serious or believing that they will resolve on their own, have been documented in several Indian and global studies [31,35]. Subsequently, self-diagnosis or/and home remedies are typical treatment patterns followed, prior to consultations with qualified or/and unqualified doctors [37,38,41]. In addition, economic factors also influence care seeking. A 2017 study from South India found income to be associated with both patient delay and provider choice [9]. In our analysis, individuals with government employment sought professional care almost a week earlier than those in agricultural occupations, and more than two weeks earlier than those who are self-employed, likely reflecting differences in income and the opportunity cost of time [42,36,43]. In terms of prior health metrics, both symptom severity and comorbidities significantly affected care-seeking behaviour, though in opposite directions: severe symptoms prompted consultation approximately five days earlier, whereas comorbidities tended to delay it, as also reported in earlier studies [13,15,40].

*"I just had a cough and did not think it would be something like TB. I thought it will go away in a few days, and used home remedies. When it did not go away for 8-9 days, I consulted with ASHA who referred me to visit a doctor in the city"* – a patient when asked why they chose home remedies

*"I followed their advice for a week but when there was no change in the symptoms, I consulted with a doctor"* – a patient who bought medication from a pharmacy, following their parents' advice

**First Provider Type & Drivers behind choosing providers.** When (professional) care is eventually sought, informal providers are the first provider choice for 30% of patients at baseline. This is aligned with earlier literature indicating that stigma, limited resources, and familiarity with local providers often shapes early care-seeking, making it difficult for the individual to pursue formal medical consultation [14,35–37]. Such patterns are particularly pronounced in high-burden, low-resource settings, where traditional and informal healthcare options are more accessible and trusted than formal facilities [13,44–46]. Consistent with earlier research [37,47,48], our baseline data shows that only 47% consulted with an allopathic doctor (at first), while 23% directly went to a pharmacy. In addition to misdiagnosis, the ease of accessing pharmacies, especially during COVID 19, was also found to be a critical factor.

"*I had cough for a couple of days and then it disappeared. I thought that it must been because of the weather change. But then a couple of months, it came back. I decided to see a doctor then*" – a patient who first sought help from an unqualified doctor

"*It is nearby, and they treat properly, everybody visits that doctor in our village, so I also went there*" – a patient who sought help from an unqualified doctor

**Diagnostic Delay & Number of Providers visited.** Reliance on unqualified practitioners, is a particular detriment to accurate diagnosis, owing to their limited diagnostic capabilities, as well as frequent misdiagnosis and delayed referral patterns, even when symptoms do not subside [4,13,37,47–50]. Many unqualified practitioners, if not all, dispense allopathic treatments such as antibiotics and injections, without formal training, or approvals [51,52]. When mired by multiple consultations and misdiagnosis with other qualified providers, it typically takes weeks or even months, for an accurate diagnosis to be reached [53,54]. Interventions such as the one assessed, have the potential to advance knowledge among unqualified providers as well, limiting unethical provider behaviour [7]. Our estimated baseline delay (27 median days) is consistent with both recent evidence (22–32 median days) [55,56], and earlier systematic reviews (31 median days) [57]. Effective referral mechanisms are therefore critical to link patients with qualified practitioners, who are better equipped to identify TB-like symptoms and initiate diagnostic testing [4,13,49,50,53,54]. Other significant determinants of diagnostic delay include comorbidities and symptom severity, which operate in opposite directions: comorbidities may mask TB symptoms and delay diagnosis [13,15,58,59], while more severe symptoms prompt earlier care-seeking.

## Impact of intervention

While no significant impact on patient delay and number of providers (visited prior to diagnosis) is observed, the intervention is associated with a significant reduction in diagnostic delay (8 + days), even after adjusting for temporal effects and demographic, clinical, and care-seeking characteristics. The results, while significant, are not consistent across the patient pool, especially those who have moderate delays. This suggests that the intervention likely mitigated extreme diagnostic lag, plausibly benefiting the most at-risk, but stronger engagement efforts are needed to bring about a reduction across the presumptive patient pool. The encouraging insight is that the intervention reduced diagnostic delay irrespective of the provider type visited by the patient [Fig 3], which suggests that formal referral linkages were instrumental in reducing time between provider visits. The same can also help reduce associated catastrophic costs, often attributed to initial alternative treatments and misdiagnosis [60,61]. Previous research has found that strengthening provider-provider and CHW-provider linkages can stimulate better use of referral services and healthcare facilities [62]. In our study, while job losses were observed to decline between phases (25% at baseline to 5%-11% at endline), distress selling remained at 20%, across the three cohorts. While we cannot be certain if reductions in diagnostic delay affected these metrics, better policies, which combine benefits across the continuum of the patient pathway, are needed to reduce the financial burdens faced by the most vulnerable [25,60,61,63]. Studies in India, and outside, find that informal providers are willing and able

to play more formal roles (referral, treatment support) if properly engaged and supported [7,64]. Combined with evidence from other Indian states, and high-burden TB regions, the intervention illustrates itself as a scalable solution to accelerate timely diagnosis, while asking for a more sustained effort if one is to bring about long term changes [45,46,53].

**COVID 19 and time effects**

Much of the baseline period (early 2021) coincided with the months following the first wave of COVID-19, whereas the endline (early 2023) reflected a period of post-pandemic normalization in healthcare services. This broader context may partly explain the reduction in both patient and diagnostic delays, which were captured as time effects in the regression model. It may also account for the greater reliance on allopathic providers observed during the endline period. Post-pandemic recovery, including the restoration of diagnostic services, heightened public awareness of respiratory illnesses, and renewed trust in formal healthcare, likely influenced these shifts in care-seeking behaviour and the reduction in the number of providers visited prior to diagnosis. These explanations are offered as post-hoc hypotheses to contextualize observed trends, and were not formally analyzed in this study. Nevertheless, they highlight how system recovery and behavioural shifts can reshape patient pathways.

**Future research**

Future studies could assess the implementation challenges of such inclusive provider engagement models, with particular focus on provider experience. Qualitative feedback from providers on their motivations and perceived barriers, especially when combined with quantitative metrics on referrals made, could inform the design of more effective interventions. In addition, estimating cost effectiveness of similar programs across diverse epidemiological and health-system contexts could guide targeted investments, not only for TB but also for other health conditions.

**Conclusion**

Our analysis demonstrates that strengthening provider linkages between formal and informal providers, supported by ease of diagnostics, can substantially reduce diagnostic delays among individuals with TB-like symptoms. Key elements, such as leveraging community and familial networks, using CXR as a cost-effective screening tool, and building trust-based referral systems, offer a scalable framework to streamline TB care pathways. The approach is replicable across other high-burden regions, both within and beyond India, and can be especially useful where informal providers are the first point of contact. Future provider engagement strategies should consciously integrate community healthcare workers and informal providers, broadening the system's reach to the very start of the patient pathway and enabling greater reductions in diagnostic delay.

**Limitations**

While comprehensive, certain limitations from the study must be acknowledged. **First**, recruitment challenges, common in qualitative field research, may have introduced some sampling bias. While most interviews were conducted at participants' homes, others were held in neutral locations such as workplaces or parks to reduce stigma and time constraints. Despite these efforts, some groups, particularly migrants and highly mobile workers, may have been underrepresented. **Second**, although interviews were designed to optimize recall of care-seeking timelines, minor inconsistencies may have affected the precision of delay estimates. However, as the DiD analysis focuses on relative changes, any recall bias is likely non-differential across periods. **Third**, the study was conducted in a predominantly urban district of India, but with significant rural and per-urban pockets. The findings, hence, may not be generalizable to regions with different healthcare infrastructure, socio-economic conditions, and cultural practices. **Fourth**, unmeasured variables may have influenced patient pathways beyond those captured by our covariates. Finally, the study period overlapped with the COVID-19 pandemic, making it difficult to fully separate intervention effects from broader post-pandemic changes in care-seeking behaviour.

## Supporting information

**S1 Appendix. Additional tables and figures consisting of robustness tests, diagnostic tests on multicollinearity.** The text also includes additional tables to support understanding relationship of comorbidities, symptom severity and occupation on patient pathway (choice of first point of care, first provider type, and mean values of patient delay & diagnostic delay).
(DOCX)

## Author contributions

**Conceptualization:** Ridhima Sodhi, Shruti Goel, Rohit Baghel, Shamim Mannan.

**Data curation:** Ridhima Sodhi, Rohit Baghel.

**Formal analysis:** Ridhima Sodhi, Shruti Goel, Kiran Rade.

**Funding acquisition:** Ridhima Sodhi, Harkesh Dabas, Shamim Mannan, Manoj Singh.

**Investigation:** Ridhima Sodhi, Kshitij Khaparde, Manoj Singh.

**Methodology:** Ridhima Sodhi, Shruti Goel, Kiran Rade, Kshitij Khaparde, Shamim Mannan.

**Project administration:** Rohit Baghel, Shamim Mannan, Manoj Singh.

**Resources:** Manoj Singh.

**Software:** Ridhima Sodhi.

**Supervision:** Harkesh Dabas, Shamim Mannan, Manoj Singh.

**Validation:** Ridhima Sodhi, Kiran Rade, Kshitij Khaparde, Harkesh Dabas, Shamim Mannan, Manoj Singh.

**Visualization:** Ridhima Sodhi.

**Writing – original draft:** Ridhima Sodhi, Shruti Goel.

**Writing – review & editing:** Ridhima Sodhi, Shruti Goel, Rohit Baghel, Kiran Rade, Kshitij Khaparde, Harkesh Dabas, Shamim Mannan, Manoj Singh.

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
