## [Decision Letter · Decision Letter 0]

14 May 2025

PGPH-D-24-02985

Strengthening TB Care Pathways: Integrating Informal and AYUSH Practitioners for Early Diagnosis and Referral

Dear Dr. Ridhima Sodhi,

Thank you for submitting your manuscript to PLOS Global Public Health. After careful consideration, we feel that it has merit but does not fully meet PLOS Global Public Health’s publication criteria as it currently stands. Therefore, we invite you to submit a revised version of the manuscript that addresses the points raised during the review process.

We look forward to receiving your revised manuscript.

Kind regards,

Alex Joseph

Academic Editor

Journal Requirements:

Additional Editor Comments (if provided):

This manuscript presents a timely and methodologically interesting quasi-experimental mixed-methods study assessing the integration of informal and AYUSH providers in the TB care pathway. However, several methodological and procedural concerns need to be addressed to strengthen the manuscript's scientific rigor and clarity.

1. Selection and Definition of Providers:

o The manuscript lacks a clear operational definition of "quacks" (unqualified providers). Also using the term “quacks” - while it is colloquially understood, it is pejorative, stigmatizing, and unprofessional in a scholarly context. How were they distinguished from AYUSH certified providers or informal allopathic practitioners? Were any formal criteria used for this classification (e.g., registration status, community nomination, verification by local health authorities)?

o What was the total provider universe, and what proportion of it was engaged in the project? The representativeness and potential selection bias in provider engagement are unclear.

2. Provider Selection and Mapping Process:

o The selection process for engaging informal and AYUSH providers is not well explained. Was snowball sampling, community health worker mapping, or an NGO supported network used?

o Were there any providers who declined to participate? What were the reasons?

o Were any systematic exclusion criteria applied to avoid engaging potentially unethical or unreliable providers?

3. Training and Standardization:

o What standardized training, if any, was provided to informal and AYUSH providers? How was their competency in identifying and referring presumptive TB patients ensured?

o Was provider adherence to referral protocols monitored throughout the intervention period?

o How was quality assurance handled, especially when involving non-certified practitioners?

4. Incentives and Risk of Misuse:

o How were financial incentives for successful referrals monitored and regulated? Was there any mechanism to prevent gaming of the system, such as unnecessary referrals?

o Did the researchers track the conversion rate of referrals to confirmed TB diagnoses by type of referring provider (AYUSH, pharmacy, quack)?

5. DiD Assumptions and Limitations:

o A key assumption of the DiD model is parallel trends. The manuscript does not provide evidence of pre-intervention trend similarity between groups.

6. Regression Model Specifications:

o The use of linear regression (OLS) for delay variables is problematic due to right-skewed distributions. Consider using GLM with appropriate link functions or quantile regression.

o The model includes many covariates relative to the sample size (n=304), raising concerns of overfitting and multicollinearity.

o Adjusted R² is moderate, and model parsimony needs better justification.

7. Sample and Response Bias:

o Non-response rate was high (~70%), and the potential for selection bias is significant. Was any weighting or adjustment done to account for this?

o Endline control and intervention groups differ in age, occupation, and comorbidities. This threatens internal validity unless more rigorous covariate balancing is applied.

8. Clustering Effects:

• Patients are nested within providers and likely within communities. No adjustments for clustering are made in regression models. This may underestimate standard errors.

9. Qualitative Component:

• The qualitative data are rich but under-integrated with quantitative findings. More structured triangulation (e.g., using joint display tables) would improve interpretation.

• The findings could be better used to explain why patient delay improved in both control and intervention arms.

10. Others

• What proportion of total presumptive TB patients referred by informal providers eventually received a TB diagnosis?

• Were there dropouts among the informal or AYUSH providers during the project? If so, what were the reasons and how were they handled?

Clarification on Ethical Review Committees: The manuscript states that ethical approvals were obtained from two different Institutional Ethics Committees for the baseline and endline assessments. For the baseline phase, ethics approval was obtained from Sigma IRB the endline phase, ethics approval was obtained from TRIOs IRB .The rationale for using two different IRBs should be clarified. It would also be helpful to confirm whether there were any protocol deviations, differences in consent forms, or interviewer procedures between the two studies due to differing IRB requirements.

Reviewers' comments:

Reviewer's Responses to Questions

**Comments to the Author**

1. Does this manuscript meet PLOS Global Public Health’s publication criteria? Is the manuscript technically sound, and do the data support the conclusions? The manuscript must describe methodologically and ethically rigorous research with conclusions that are appropriately drawn based on the data presented.? Is the manuscript technically sound, and do the data support the conclusions? The manuscript must describe methodologically and ethically rigorous research with conclusions that are appropriately drawn based on the data presented.

Reviewer #1: Yes

2. Has the statistical analysis been performed appropriately and rigorously?

Reviewer #1: Yes

3. Have the authors made all data underlying the findings in their manuscript fully available (please refer to the Data Availability Statement at the start of the manuscript PDF file)?

The PLOS Data policy requires authors to make all data underlying the findings described in their manuscript fully available without restriction, with rare exception. The data should be provided as part of the manuscript or its supporting information, or deposited to a public repository. For example, in addition to summary statistics, the data points behind means, medians and variance measures should be available. If there are restrictions on publicly sharing data—e.g. participant privacy or use of data from a third party—those must be specified.requires authors to make all data underlying the findings described in their manuscript fully available without restriction, with rare exception. The data should be provided as part of the manuscript or its supporting information, or deposited to a public repository. For example, in addition to summary statistics, the data points behind means, medians and variance measures should be available. If there are restrictions on publicly sharing data—e.g. participant privacy or use of data from a third party—those must be specified.

Reviewer #1: No

4. Is the manuscript presented in an intelligible fashion and written in standard English?

Reviewer #1: Yes

Reviewer #1: Well presented manuscript, the methodology was good and the analysis precise and clear. Data availability statement has not been added.

Is there a reason why weighting or sensitivity analysis was not done to control for high non-response rate?

**Do you want your identity to be public for this peer review?** For information about this choice, including consent withdrawal, please see our Privacy Policy..

Reviewer #1: No

---

## [Decision Letter · Decision Letter 1]

17 Sep 2025

PGPH-D-24-02985R1

Strengthening TB Care Pathways: Integrating Informal and AYUSH Practitioners for Early Diagnosis and Referral

Dear Ridhima Sodhi,

Thank you for submitting your manuscript to PLOS Global Public Health. After careful consideration, we feel that it has merit but does not fully meet PLOS Global Public Health’s publication criteria as it currently stands. Therefore, we invite you to submit a revised version of the manuscript that addresses the points raised during the review process.

Please submit your revised manuscript by Sepetember 30, If you will need more time than this to complete your revisions, please reply to this message or contact the journal office at globalpubhealth@plos.org. Please include the following items when submitting your revised manuscript:

We look forward to receiving your revised manuscript.

Kind regards,

Alex Joseph

Academic Editor

Journal Requirements:

Reviewers' comments:

Reviewer's Responses to Questions

**Comments to the Author**

Reviewer #2: (No Response)

Reviewer #3: (No Response)

publication criteria? Is the manuscript technically sound, and do the data support the conclusions? The manuscript must describe methodologically and ethically rigorous research with conclusions that are appropriately drawn based on the data presented.? Is the manuscript technically sound, and do the data support the conclusions? The manuscript must describe methodologically and ethically rigorous research with conclusions that are appropriately drawn based on the data presented.

Reviewer #2: Partly

Reviewer #3: No

3. Has the statistical analysis been performed appropriately and rigorously?

Reviewer #2: Yes

Reviewer #3: No

4. Have the authors made all data underlying the findings in their manuscript fully available (please refer to the Data Availability Statement at the start of the manuscript PDF file)?

The PLOS Data policy requires authors to make all data underlying the findings described in their manuscript fully available without restriction, with rare exception. The data should be provided as part of the manuscript or its supporting information, or deposited to a public repository. For example, in addition to summary statistics, the data points behind means, medians and variance measures should be available. If there are restrictions on publicly sharing data—e.g. participant privacy or use of data from a third party—those must be specified.requires authors to make all data underlying the findings described in their manuscript fully available without restriction, with rare exception. The data should be provided as part of the manuscript or its supporting information, or deposited to a public repository. For example, in addition to summary statistics, the data points behind means, medians and variance measures should be available. If there are restrictions on publicly sharing data—e.g. participant privacy or use of data from a third party—those must be specified.

Reviewer #2: Yes

Reviewer #3: Yes

5. Is the manuscript presented in an intelligible fashion and written in standard English?

Reviewer #2: Yes

Reviewer #3: Yes

Reviewer #2: Dear authors,

While you've made meaningful improvements to the statistical methods and presentation, several critical issues limit the manuscript's contribution:

Major concerns:

- The 70% non-response rate creates substantial selection bias that stratified sampling cannot address. Your dismissal of sensitivity analyses is unconvincing given this magnitude of non-response threatens generalizability.

- Only 35% of mapped providers were "actively engaged" - much lower than the engagement narrative suggests. This raises questions about intervention fidelity.

- Low TB diagnosis rates from referrals (5-13%) suggest potential over-referral issues that warrant discussion.

Moderate concerns:

- It is unclear whether "non-intervention" endline group had any exposure to intervention components.

- There is high provider dropout rates acknowledged but not systematically analyzed.

Minor concerns:

- Use clear descriptive labels instead of the variable names in table 3.

- Some sentences are overly long and could be broken down for clarity (particularly in Methods section)

Recommendations:

- Conduct sensitivity analyses or inverse probability weighting to address non-response bias

- Provide honest assessment of intervention uptake (35% active engagement)

- Discuss implications of low referral conversion rates

- Clarify control group definition and potential contamination

The study addresses important questions but methodological limitations significantly constrain the strength of conclusions.

Reviewer #3: This manuscript addresses an important and underexplored issue and the topic is highly relevant for TB-affected countries like India, where informal providers are often the first point of contact. The mixed-methods quasi-experimental design and use of a Differences-in-Differences (DiD) analysis are strengths. The findings, particularly the reduction in diagnostic delay could make a valuable contribution to global TB care literature.

However, there are substantial methodological, interpretive, and reporting concerns that must be addressed before publication. In its current form, the manuscript risks overstating causality, underexplaining bias, and presenting findings that may not be generalisable.

1. Methodological Rigour

Sampling bias: A 60–70% refusal rate introduces serious non-response bias. The sample may disproportionately reflect individuals with lower stigma or greater availability, undermining representativeness.

Baseline imbalance: At baseline, no qualified providers were recorded as first point of contact; this is attributed to differences in field staff training. This undermines comparability between baseline and endline groups, a critical assumption of DiD.

Confounding (COVID-19 effect): The intervention period coincided with the COVID-19 pandemic. Changes in care-seeking behaviour, mobility, and health infrastructure may independently explain the improvements. The paper acknowledges this but does not convincingly disentangle intervention effects from pandemic-related shifts.

Outcome measurement: Patient and diagnostic delays rely on patient recall. This is highly prone to bias and may not be reliable, especially with long recall periods.

2. Analysis and Interpretation

The DiD models include multiple covariates but are not presented in a way that allows readers to assess robustness. Adjusted R² values are modest, and the text overemphasises the intervention’s effect while underplaying secular time trends.

Patient delay is reported to decline by 19 days overall, but the paper admits this is not attributable to the intervention. Yet the discussion blurs this distinction, overstating the intervention’s role.

Reduction in number of providers visited is attributed to improved pathways, but this may equally reflect pandemic-related service disruption or differences in symptom severity.

3. External Validity

Findings are from one rural district in Chhattisgarh, India, and cannot be assumed to generalise to other Indian states or other high-burden countries. The conclusion makes overbroad claims about scalability.

4. Ethical Concerns

The intervention incentivised informal providers to act as treatment supporters. This raises ethical issues about legitimising unqualified providers without adequate oversight. The discussion should critically engage with the risks of formalising informal practice.

5. Writing and Structure

The manuscript is overly long, with repetition across abstract, results, and discussion. It could be tightened considerably.

Figures and tables are dense; clearer visual presentation would improve readability.

References are dated in places; more recent TB patient pathway studies (post-2021) should be cited.

**Do you want your identity to be public for this peer review?** For information about this choice, including consent withdrawal, please see our Privacy Policy..

Reviewer #2: No

Reviewer #3: No

---

## [Decision Letter · Decision Letter 2]

7 Jan 2026

PGPH-D-24-02985R2

Strengthening TB Care Pathways: Integrating Informal and AYUSH Practitioners for Early Diagnosis and Referral

Dear Dr. Sodhi,

Thank you for submitting your manuscript to PLOS Global Public Health. After careful consideration, we feel that it has merit but does not fully meet PLOS Global Public Health’s publication criteria as it currently stands. Therefore, we invite you to submit a revised version of the manuscript that addresses the points raised during the review process.

We look forward to receiving your revised manuscript.

Kind regards,

Alex Joseph

Academic Editor

Journal Requirements:

Additional Editor Comments (if provided):

Reviewers' comments:

Reviewer's Responses to Questions

**Comments to the Author**

Reviewer #4: (No Response)

Reviewer #5: (No Response)

publication criteria? Is the manuscript technically sound, and do the data support the conclusions? The manuscript must describe methodologically and ethically rigorous research with conclusions that are appropriately drawn based on the data presented.? Is the manuscript technically sound, and do the data support the conclusions? The manuscript must describe methodologically and ethically rigorous research with conclusions that are appropriately drawn based on the data presented.

Reviewer #4: Yes

Reviewer #5: Yes

3. Has the statistical analysis been performed appropriately and rigorously?

Reviewer #4: Yes

Reviewer #5: Yes

4. Have the authors made all data underlying the findings in their manuscript fully available (please refer to the Data Availability Statement at the start of the manuscript PDF file)?

The PLOS Data policy requires authors to make all data underlying the findings described in their manuscript fully available without restriction, with rare exception. The data should be provided as part of the manuscript or its supporting information, or deposited to a public repository. For example, in addition to summary statistics, the data points behind means, medians and variance measures should be available. If there are restrictions on publicly sharing data—e.g. participant privacy or use of data from a third party—those must be specified.requires authors to make all data underlying the findings described in their manuscript fully available without restriction, with rare exception. The data should be provided as part of the manuscript or its supporting information, or deposited to a public repository. For example, in addition to summary statistics, the data points behind means, medians and variance measures should be available. If there are restrictions on publicly sharing data—e.g. participant privacy or use of data from a third party—those must be specified.

Reviewer #4: Yes

Reviewer #5: Yes

5. Is the manuscript presented in an intelligible fashion and written in standard English?

Reviewer #4: Yes

Reviewer #5: Yes

Reviewer #4: A good study showing how integration of informal and AYUSH practitioners helps in early diagnosis and referral of tuberculosis. Similar approach was practiced in other diseases like kala-azar as well. Having said this, a word of caution is also required to limit informal and AYUSH practitioners to referral only, and not towards treatment modalities other than ATT. During implementation, local practitioners should be strictly advised against indiscriminate use of antibiotics, steroids or other AYUSH treatments before or after diagnosis.

Reviewer #5: I am reviewing Revision 2. I have not seen the previous versions of this manuscript, nor have I submitted any previous suggested revisions. I do not have access to the reviewer comments from previous reviewers, but there was clearly a significant revision in the manuscript, based in previous reviewers comments.

In addition, Revision 2 does not have manuscript line numbering, so I can only refer to page numbers for suggested changes.

This manuscript describes a "quasi-experimental" study that evaluates the impact of an intervention that links non-allopathic health providers, including traditional healers and AYUSH practitioners, with the formal allopathic medical community to improve the diagnosis and treatment of pulmonary tuberculosis. The final conclusions highlight that there was a significant reduction in the delay in diagnosis of TB due to the intervention.

GENERAL COMMENTS: The authors do describe the intervention with non formal traditional providers but it would be helpful to have much more detail. How long did this intervention take? (weeks, months??) How much did it cost? How were the various target audiences of non-allopathic "healers" identified and who provided the information about referrals, access to CXR, etc? Was feedback provided to these traditional healers after the study was completed? Did the non-allopathic healers feel positive about this interaction and did they have any useful suggestions for future action? What would the authors have done differently? This would be helpful for readers who are contemplating a similar exercise in their own setting.

SPECIFIC COMMENTS:

Pages 1-2. I would suggest removing the AUTHOR SUMMARY. It is duplicative of the Abstract and the Introduction.

Page 2, INTRODUCTION: Fourth sentence, replace, "...well-known..." with "...common..."

Page 2 INTRODUCTION-8th sentence, it's not clear what the phrase, "...the same continues to...."means. Can you rewrite this?

Page 2 INTRODUCTION, Line 10, add "where" after the word "region."

Page 2, INTRODUCTION, Line 11, can you state what this proportion of 4.4% and 1.4% refers to. Is this of the Total population of health seeking persons?

Page 3, METHODS, What is the meaning of "DS" in the subtitle?

Page 8, RESULTS, Demographic survey,. The authors state that the majority of patients in this study are male. Would they comment on possible differences if there were more female subjects involved?

Page 16 TABLE 6. This was a very useful and interesting table. Congratulations

Page 17, under "PATIENT DELAY......." Line 3 should be revised to read "......visit a medical professional based on (or not) advice given by close contacts....."

**Do you want your identity to be public for this peer review?** For information about this choice, including consent withdrawal, please see our Privacy Policy..

Reviewer #4: **Yes:** Dr. Madhuri DevarajuDr. Madhuri DevarajuDr. Madhuri DevarajuDr. Madhuri Devaraju

Reviewer #5: **Yes:** Paul R De Lay, MD, DTM&H (Lond)Paul R De Lay, MD, DTM&H (Lond)Paul R De Lay, MD, DTM&H (Lond)Paul R De Lay, MD, DTM&H (Lond)

---

## [Decision Letter · Decision Letter 3]

17 Mar 2026

Strengthening TB Care Pathways: Integrating Informal and AYUSH Practitioners for Early Diagnosis and Referral

PGPH-D-24-02985R3

Dear Sodhi,

We are pleased to inform you that your manuscript 'Strengthening TB Care Pathways: Integrating Informal and AYUSH Practitioners for Early Diagnosis and Referral' has been provisionally accepted for publication in PLOS Global Public Health.

Best regards,

Julia Robinson

Executive Editor

Reviewer Comments (if any, and for reference):

Reviewer's Responses to Questions

**Comments to the Author**

Reviewer #5: All comments have been addressed

publication criteria? Is the manuscript technically sound, and do the data support the conclusions? The manuscript must describe methodologically and ethically rigorous research with conclusions that are appropriately drawn based on the data presented.? Is the manuscript technically sound, and do the data support the conclusions? The manuscript must describe methodologically and ethically rigorous research with conclusions that are appropriately drawn based on the data presented.

Reviewer #5: Yes

3. Has the statistical analysis been performed appropriately and rigorously?

Reviewer #5: Yes

4. Have the authors made all data underlying the findings in their manuscript fully available (please refer to the Data Availability Statement at the start of the manuscript PDF file)?

The PLOS Data policy requires authors to make all data underlying the findings described in their manuscript fully available without restriction, with rare exception. The data should be provided as part of the manuscript or its supporting information, or deposited to a public repository. For example, in addition to summary statistics, the data points behind means, medians and variance measures should be available. If there are restrictions on publicly sharing data—e.g. participant privacy or use of data from a third party—those must be specified.requires authors to make all data underlying the findings described in their manuscript fully available without restriction, with rare exception. The data should be provided as part of the manuscript or its supporting information, or deposited to a public repository. For example, in addition to summary statistics, the data points behind means, medians and variance measures should be available. If there are restrictions on publicly sharing data—e.g. participant privacy or use of data from a third party—those must be specified.

Reviewer #5: Yes

5. Is the manuscript presented in an intelligible fashion and written in standard English?

Reviewer #5: Yes

Reviewer #5: This is my second review of the manuscript. The authors have done an excellent job in responding to my comments and suggestions. They have adequately explained the rationale for not making a recommended change and have made critical changes based on my suggestions, for example in the "Future Research" section. These revisions should make this paper more useful for other TB programs throughout the world to explore and assess the possibly critical role of non-allopathic providers in accelerating and improving TB diagnosis and treatment.

SPECIFIC SUGGESTED REVISION:

Page 3, middle of the page, please spell out the meaning of "ICT"j since this is the first use of the abbreviation.

**Do you want your identity to be public for this peer review?** For information about this choice, including consent withdrawal, please see our Privacy Policy..

Reviewer #5: **Yes:** Paul R De Lay, MD, DTM&H (Lond)Paul R De Lay, MD, DTM&H (Lond)Paul R De Lay, MD, DTM&H (Lond)Paul R De Lay, MD, DTM&H (Lond)
